# Detectable Vancomycin Stool Concentrations in Hospitalized Patients with Diarrhea Given Intravenous Vancomycin

Taryn A. Eubank [ID], Chenlin Hu [ID], Anne J. Gonzales-Luna [ID] and Kevin W. Garey *[ID]

Department of Pharmacy Practice and Translational Research, College of Pharmacy, University of Houston, Houston, TX 77204, USA; taeubank@central.uh.edu (T.A.E.); chu10@central.uh.edu (C.H.); ajgonz23@central.uh.edu (A.J.G.-L.)

* Correspondence: kgarey@central.uh.edu

**Abstract:** Vancomycin is not appreciably passaged via the colonic membrane to the gastrointestinal (GI) tract in persons with an intact gut epithelium due to its large chemical structure. However; hospitalized patients with diarrhea often have a disrupted GI tract. The aim of this study was to determine the frequency of detectable vancomycin concentrations in the stool of patients with antibiotic-associated diarrhea receiving IV vancomycin. This was a multicenter cohort study of hospitalized patients with stool samples collected for *Clostridioides difficile* testing. Leftover stool samples were collected from patients who had received least 3 days of IV vancomycin. Fecal vancomycin was quantified by high-performance liquid chromatography. The study cohort included 33 unique patients, majority female (54.5%) aged 60 years (range 23–84). Eighteen of thirty-three patients (54.5%) tested positive for *C. difficile* toxins. The average duration of systemic vancomycin administration prior to stool collection was 3.5 (range 2–15) days. Three of 33 (9%) stool samples had a detectable vancomycin concentration (range 1.2–13.2 mcg/mL). These concentrations may promote the development of vancomycin-resistant *Enterococcus* or van mutations in *C. difficile*, leading to vancomycin resistance. Further studies on implications are warranted.

**Keywords:** glycopeptides; stool; pharmacokinetics; *Clostridioides difficile*; microbiome

## 1. Introduction

The glycopeptide antibiotic vancomycin is one of the most used antimicrobials in the United States given orally for *Clostridioides difficile* infection (CDI) and intravenously (IV) for systemic Gram-positive infections [1–3]. As a water-soluble, tricyclic glycosylated peptide, oral vancomycin cannot be stepped down from the IV formulation as it is too large to cross the intestinal barrier in meaningful amounts [4–7]. Similarly, although oral vancomycin causes severe dysbiosis to the gut microbiome [8], IV vancomycin is considered a low-risk antibiotic due to lack of colonic concentrations. However, several case series have described systemic levels of vancomycin following oral administration [9–13], indicating an ability to cross the intestinal mucosal barrier in special populations. Inversely, systemic or intravenous (IV) vancomycin entering the GI tract has only been briefly described in two publications. Geraci et al. in 1956 described vancomycin stool concentrations following intravenous administration in six patients and 33 patients were included in a study by Currie and Lemos-Filho in 2004 [14,15]. Currie and Lemos-Filho described vancomycin concentrations in bile samples using an AxSym II fluorescence polarization immuno-assay from 2 of the 33 patients as a possible venue for entering the GI tract. However, this study excluded other variables that have been documented as risk factors for systemic absorption of oral vancomycin such as renal insufficiency (creatinine clearance [CrCl] < 50 mL/min) and evidence of active bowel pathology (diarrhea, inflammation, and malignancy). Systemic inflammation diminishes the integrity of the GI mucosa, allowing for translocation of larger molecules such as vancomycin [13,16,17]. Bowel pathology, inflammation, and/or

diarrhea have also been described as risk factors for vancomycin gut penetration due to increased intestinal permeability [16,17]. However, fecal vancomycin clearance is increased in patients with profuse diarrhea [5], potentially reducing concentrations in the stool due to a dilution effect. Here, we sought to analyze the frequency and levels of IV vancomycin in the stool of patients with active diarrhea to further elucidate gut microbiome exposure following IV vancomycin administration.

## 2. Results

### 2.1. Population Demographics

The study cohort was comprised of a single stool sample from each of 33 individual patients. The majority were female (54.5%) and the mean age in years was 59.6. (Table 1). Active infection was present in 84.8% of the population with 18/33 (54.5%) patients being diagnosed with CDI. The average days of systemic vancomycin administration prior to stool collection was 3.5 days (range 2–15).

**Table 1.** Patient demographics.

| Characteristic | Overall (*n* = 33) | Vancomycin Detected in Stool (*n* = 3) | No Vancomycin in Stool (*n* = 30) |
|---|---|---|---|
| Male sex | 15 (45.5%) | 3 (100%) | 12 (40%) |
| Age (years, mean (range)) | 60 (23–84) | 62 (23–84) | 59 (56–71) |
| Serum creatinine * (mg/dL, mean (range)) | 1.45 (0.4–9.49) | 1.42 (0.51–2.7) | 1.45 (0.4–9.49) |
| Receiving renal replacement (HD/PD) | 5 (15.2%) | 1 (33.3%) | 4 (13.3%) |
| Intensive Care Unit [a] | 16 (48.5%) | 3 (100%) | 13 (43.3%) |
| Active infection | 28 (84.8%) | 2 (66.7%) | 26 (86.7%) |
|   Intrabdominal infection | 4 (12.1%) | 2 (66.7%) | 2 (6.67%) |
|   CDI | 18 (54.5%) | 2 (66.7%) | 16 (53.3%) |
|   Septic shock | 10 (30.3%) | 0 (0%) | 10 (33.3%) |
| GI bleed w/in 1 month | 3 (9.1%) | 0 (0%) | 3 (10%) |
| Diagnosis of IBS | 0 (0.0%) | -- | -- |
| Diagnosis of diverticulitis | 0 (0.0%) | -- | -- |
| Diagnosis of Crohn's disease | 0 (0.0%) | -- | -- |
| History of bariatric surgery | 2 (6.1%) | 0 (0%) | 2 (6.67%) |
| GI surgery w/in 1 month | 5 (15.2%) | 0 (0%) | 5 (16.7%) |
| Receiving tube feeds | 12 (36.4%) | 3 (100%) | 9 (30%) |
| Immunocompromised [¥] | 12 (36.4%) | 1 (33.3%) | 11 (36.7%) |
| Active malignancy | 7 (21.2%) | 1 (33.3%) | 6 (20%) |
| Days of vancomycin (mean (range)) | 3.54 (2–15) | 2.67 (2–4) | 3.63 (2–15) |
| Vancomycin dose (mg/kg/dose, mean (range)) | 14.5 (10.2–26) | 11.9 (10.5–12.8) | 14.8 (10.2–26) |
| Vancomycin serum trough level (mcg/mL, mean (range)) ** | 14.7 (6.0–24.3) | 14.4 (6.0–19.8) | 14.8 (7.8–24.3) |

* Within 24 h of stool collection; [a] at time of stool collection; [¥] immunocompromised includes receipt of $\geq$20 mg per day of prednisone equivalent for at least 2 weeks, antineoplastic therapy, tumor necrosis factor alpha inhibitor, or a calcineurin inhibitor during their hospitalization stay; ** within 48 h of stool collection, *n* = 30/33; Abbv.: HD/PD, hemodialysis/peritoneal dialysis; CDI, *C. difficile* infection; GI, gastrointestinal; IBS, irritable bowel syndrome. Numbers are value (%) unless otherwise stated.

### 2.2. High-Performance Liquid Chromatography (HPLC) Results

Approximately 9% (3/33) of samples had a detectable vancomycin level. The respective patients were all male sex and the range of detection was 1.2 mcg/mL to 13.2 mcg/mL. Additional patient characteristics and risk factors will be discussed and can be found in Table 2.

### 2.3. Patient 1

Patient 1 is a 56-year-old male with a vancomycin stool level of 1.2 mcg/mL. Vancomycin was administered for 48 h prior to stool collection. On the day of stool collection (day 1), a vancomycin serum level was collected and resulted in 16 mcg/mL. An additional level was drawn on day 2 that resulted at 19.8 mcg/mL. The patient had a past medical history significant for end-stage renal disease on hemodialysis, chronic respiratory failure with chronic mechanical ventilation via tracheostomy, and diabetes. The current admission was due to fever (101 °F). The patient had been on long-term tube feeds prior to admission. Significant microbiology reports for this admission include blood cultures

positive for methicillin-resistant Staphylococcus aureus (MRSA) and the stool was positive for *C. difficile*.

**Table 2.** Notable characteristics of patients with detected stool vancomycin concentrations.

| Patient Characteristics | Patient 1 | Patient 2 | Patient 3 |
|---|---|---|---|
| Vancomycin level (mcg/mL) | 1.2 | 4.8 | 13.2 |
| Male sex | Yes | Yes | Yes |
| Age (years) | 56 | 71 | 59 |
| Serum creatinine (mg/dL, mean) * | 2.7 | 1.04 | 0.51 |
| Receiving renal replacement (HD/PD) | Yes | No | No |
| Intensive care unit [EUR] | Yes | Yes | Yes |
| Active infection | Yes | No | Yes |
|    Intrabdominal | Yes | No | Yes |
|    CDI | Yes | No | Yes |
|    Septic shock | No | No | No |
| Receiving tube feeds | Yes | Yes | Yes |
| Immunocompromised | No | No | Yes |
| Active Malignancy | No | Yes | No |
| Days of prior vancomycin administration | 2 | 2 | 4 |
| Vancomycin dose (mg/kg, mean) | 10.5 | 12.8 | 12.4 |
| Vancomycin serum trough level (mcg/mL) ** | 19.8 | 17.4 | 6 |

* Within 24 h of stool collection; [EUR] at time of stool collection; ** within 48 h of stool collection; Abbv.: HD/PD, hemodialysis/peritoneal dialysis; CDI, *C. difficile* infection; GI, gastrointestinal; IBS, irritable bowel syndrome.

### 2.4. Patient 2

Patient 2 is a 71-year-old male with a vancomycin stool level of 4.8 mcg/mL. Two days of vancomycin were administered prior to stool collection with a vancomycin serum level of 17.4 mcg/mL on day 3. Past medical history is significant for recent diagnosis of hepatitis C and right maxillary small cell carcinoma. The current admission was for maxillectomy, neck dissection, and right lower extremity free flap with cheek reconstruction (8 days before stool collection). Tube feeds were initiated post-operation for nutrition support. No active infection was documented from culture results; antibiotics were administered for post-operation infection prophylaxis.

### 2.5. Patient 3

The last patient is a 59-year-old male with a vancomycin stool level of 13.2 mcg/mL. The patient received 4 days of vancomycin prior to stool collection with serum levels of 6 mcg/mL and 17.1 mcg/mL on day 1 and day 4, respectively. The past medical history is significant for idiopathic pulmonary fibrosis status post double lung transplant 4 days prior to stool collection. The admission was complicated by hemodynamic instability and vasoplegic shock post-transplant, necessitating pressor support. Tube feeds were initiated post-transplant for nutrition support. Post-transplant bronchoscopy cultures grew *Mycobacterium chelonae*, *Mycobacterium mucogenicum*, and MRSA. Additionally, the stool was positive for *C. difficile*. Renal function during admission was stable and within normal limits.

### 3. Discussion

Systemic detection of vancomycin following oral administration has been reported in several studies [9–13], while IV vancomycin detected in the stool has only been described twice [7,14]. Both studies previously describing IV vancomycin gut penetration have excluded patients with risk factors for GI penetration, including bowel inflammation/diarrhea, reduced renal function, and/or high, prolonged vancomycin dosing [7,14]. The first only tested stool samples from a random sample of eight healthy volunteers receiving IV vancomycin and reported fecal concentrations ranging from 0–110 mg/L [7]. The second noted detectable fecal concentrations in 19% (5/26) and 95% (26/28) of samples

collected <5 days into therapy and ≥5 days of therapy, respectively [14]. Overall, stool levels of vancomycin ranged from <2–94.8 mg/L.

We present three patient cases with detectable vancomycin concentrations in the stool out of 33 screened patients. We included patients with diarrhea warranting *C. difficile* testing to assess bowel penetration in this high-risk population. The overall rate of detectable fecal vancomycin in our study was comparable to previous reports: 9.1% (3/33) versus 19.2% (5/26) [14]. However, we did not note a higher rate of vancomycin bowel penetration in those with other previously described risk factors, including renal replacement therapy, longer durations of vancomycin, or higher doses of vancomycin. As fecal vancomycin was only detectable in patients following 4 days or less of IV therapy, it seems likely inflammation or diarrheal osmosis may be contributing to our observations. These findings indicate vancomycin bowel penetration may be multifactorial.

Our study has several unique strengths that differ from the previous literature [7,14]. First, patients with active bowel pathology or inflammatory states were included, as were. those with renal insufficiency. Additionally, the duration of systemic vancomycin administration prior to stool collection was shorter than in previous studies which reported the majority of stool to only have detectable levels after 5 days of therapy [14]. The risk factors associated with oral vancomycin administration leading to systemic absorption are better described than the reverse phenomenon, presented here. Repeatedly described risk factors include bowel pathology leading to inflammation and/or diarrhea or accumulation of vancomycin via reduced renal function or prolonged high doses of vancomycin [9–13]. Our study supports systemic vancomycin bowel penetration to possibly be multifactorial. The three patients each had diarrhea due to suspected or confirmed CDI in addition to receipt of tube feeds, which can additionally cause diarrhea. Additional risk factors observed in the individual patients were renal insufficiency, other active infections, active malignancy, and systemic shock. Infections, specifically bloodstream infections, malignancy, and systemic shock can all place the individual in a state of inflammation. Severe inflammation can diminish the integrity of the GI mucosa allowing the absorption of larger molecules like vancomycin. Interestingly, our study does not support the hypothesis that systemic accumulation due to high or prolonged dosing is the main risk factor driving bowel penetration. Although Patient 1 had renal insufficiency, serum vancomycin levels were not supratherapeutic (>20 mg/L). Additionally, our patients had vancomycin concentrations in the stool at day 4 or sooner, leading to the hypothesis that inflammation and diarrhea may be the driving factor behind our observed bowel penetration rather than accumulation.

The analysis in our study is limited by the small number of patients testing positive for fecal vancomycin and heterogeneity associated with our clinically relevant population of hospitalized patients with diarrhea. Additionally, some stool samples were stored long-term (3–4 years) at −80 °C prior to HPLC analysis. It is possible, but unlikely, that vancomycin may have degraded in the sample, leading to an underestimation of true concentrations.

## 4. Conclusions

We report that 9% of patients with antibiotic-associated diarrhea have detectable levels of vancomycin in their stool. Implications of low levels of vancomycin on the gut microbiome and resistome remain largely unknown. However, as these patients likely have underlying dysbiosis associated with antibiotic receipt, vancomycin bowel penetration carries a concern for resistance development in Enterococcus spp. and/or *C. difficile* [18–22]. Further studies on the prevalence of IV vancomycin bowel penetration and its effect on the microbiome and resistome are warranted.

## 5. Materials and Methods

We conducted a multicenter, retrospective study from two hospital systems in the Texas Medical Center, Houston, Texas. Leftover stool samples from adult patients tested for CDI as part of routine clinical care were collected between 2016–2019 and brought to the research

laboratory at the University of Houston. Patients' electronic health records were screened for (1) IV vancomycin receipt for $\geq$48 h prior to stool collection, (2) $\geq$1 dose of IV vancomycin administered <24 h prior to stool collection (exception for hemodialysis patients with documented serum level), and (3) no oral vancomycin administration prior to stool collection. Factors associated with increased risk of vancomycin bowel penetration were collected and included renal function/renal replacement therapy, active bowel pathology (irritable bowel syndrome (IBS), diverticulitis, Crohn's disease, history of bariatric surgery, GI surgery within the last month, GI bleed within the last month), and active inflammatory states (active bowel pathology, active infection, shock as evidenced by vasopressor support, and active malignancy) [15,17,23,24]. This study was approved by the Committee for the Protection of Research Subjects at the University of Houston (CPHS000128). This analysis follows STROBE (Strengthening the Reporting of Observational Studies in Epidemiology) Recommendations [25].

Vancomycin stool concentrations were measured using HPLC as previously described [26]. Briefly, an aliquot of stool and 1 mL of extraction solvent (acetonitrile and water, 10:90 (*v/v*), 0.1% TFA) were added into a 2 mL centrifuge tube and shaken (Vortex-Genie 2 Shaker, Scientific Industries, Bohemia, NY, USA) for 10 seconds, ultrasonicated for 5 min (Multool ultrasonic cleaner, Model: TH-030A), and centrifuged at $10,000\times g$ for 3 min (Centrifuge 5804 R, Eppendorf, Hamburg, Germany). The supernatant was collected and diluted 10-fold using the extraction solvent and was stored at $-80\ ^{\circ}$C prior to HPLC analysis. Fecal vancomycin was quantified based on the standard calibration curve with vancomycin standard concentrations plotted against the corresponding HPLC peak areas. Results were reported as descriptive statistics.

**Author Contributions:** Methodology and writing—original draft preparation, T.A.E.; methodology and writing—review and editing, C.H.; writing—review and editing, A.J.G.-L.; writing—review and editing, K.W.G. All authors have read and agreed to the published version of the manuscript.

**Funding:** This research was supported in part by the NIH National Institute of Allergy and Infectious Diseases (NIAID) T32 AI141349; NIAID R01AI139261; ACCP Foundation Junior Investigator Research Award: G0507743; SIDP Early Career Research Award: G0507861.

**Institutional Review Board Statement:** This study was approved by the Committee for the Protection of Research Subjects at the University of Houston (CPHS000128).

**Informed Consent Statement:** Not applicable.

**Data Availability Statement:** The data presented in this study are available on request from the corresponding author.

**Conflicts of Interest:** K.W.G. has research grants from Summit Therapeutics and Acurx Pharmaceuticals. All other authors have no reported conflict of interest.

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
