# Peer review of "Detectable Vancomycin Stool Concentrations in Hospitalized Patients with Diarrhea Given Intravenous Vancomycin"

_2813-0618, doi:10.3390/pharma2040024_

Round 1

Reviewer 1 Report

Major Concerns:

1. The study lacks a clear rationale for assessing vancomycin levels in patients. The introduction section should be revised to provide a comprehensive background with appropriate references explaining why it is important to study vancomycin levels in patients.

2. The proportion of positive samples in this study appears to be very low. This raises doubts about the statistical significance of the results. The authors should consider expanding the sample size or addressing this limitation in the discussion section.

3. It is unclear which part of the study is retrospective and which is prospective. To enhance clarity, the authors should use a flowchart or flow diagram to differentiate between these aspects.

4. The methodology section needs improvement to align with STROBE guidelines. Specifically, the study duration should be clearly stated to provide a better understanding of the data collection process.

5. The discussion section lacks a comparison between the group with 'no detected vancomycin' and the group with 'detected vancomycin levels'. It is essential to include this comparison and provide justifications for any differences observed.

Minor Concerns:

1. The first sentence of the results section needs clarification. It appears to suggest that 33 samples were taken from individual patients. This sentence should be revised for clarity to accurately convey the sample size and methodology.

Author Response

Reviewer #1

Major Concerns:

  1. The study lacks a clear rationale for assessing vancomycin levels in patients. The introduction section should be revised to provide a comprehensive background with appropriate references explaining why it is important to study vancomycin levels in patients.

Inclusion of comparison to other studies that omitted clinically relevant patients who have renal replacement requirements or systemic inflammation ongoing. Additionally, verbiage added on gut microbiome exposure increase if IV vancomycin is penetrating the GI in these clinical settings possibly leading to antimicrobial resistance generation and gut microbiome dysbiosis.

  1. The proportion of positive samples in this study appears to be very low. This raises doubts about the statistical significance of the results. The authors should consider expanding the sample size or addressing this limitation in the discussion section.

 The authorship team agrees and does acknowledge the small size is a limitation and is why descriptive statistics were performed (see previous acknowledgment on line 127: “Analysis in our study is limited by the small number of patients testing positive for fecal vancomycin and heterogeneity associated with our clinically relevant population of hospitalized patients with diarrhea.” However, this is the largest population to be analyzed thus far and we believe it does contribute proof-of-concept for future larger studies. Future work to see the implication of IV vancomycin penetrating the GI such as antimicrobial resistance generation or dysbiosis of the microbiome would be of great interest and could stem from this publication being available.

  1. It is unclear which part of the study is retrospective and which is prospective. To enhance clarity, the authors should use a flowchart or flow diagram to differentiate between these aspects.

 This is translational research on samples from our biorepository making the lines between retrospective and prospective grey. These samples are historic samples and chart review must be completed making it retrospective in nature. Will defer to editor if further clarification is warranted.

  1. The methodology section needs improvement to align with STROBE guidelines. Specifically, the study duration should be clearly stated to provide a better understanding of the data collection process.

Thank you for your comment and pointing out the timeframe had been left out. Materials and Methods section updated to include patient sample collection timeframe.

  1. The discussion section lacks a comparison between the group with 'no detected vancomycin' and the group with 'detected vancomycin levels'. It is essential to include this comparison and provide justifications for any differences observed.

Discussion updated and expanded upon.

Minor Concerns:

  1. The first sentence of the results section needs clarification. It appears to suggest that 33 samples were taken from individual patients. This sentence should be revised for clarity to accurately convey the sample size and methodology.

The first sentence has been modified to reflect 1 singular stool sample was taken from 33 individual patients. Now reading: “The study cohort was comprised of a singular stool sample from 33 individual patients.”

Reviewer 2 Report

-Line 24 reads “Three of 33 (9%) samples had a detectable vancomycin concentration (range 1.2-13.2 mcg/mL)”. Please, indicate that these values correspond to FECAL samples and indicate the mean/median time to which these levels correspond with respect to the beginning of IV vancomycin treatment.

 -Line 50: please indicate to which number of participants the samples correspond. I understand that there were 33 different patients, but this should be made explicit in the text.

-Lines 134-135: please add the information related to the setting from which patients were selected (medical, surgical or both), the age range of the participants (>18 years), and the study period in which patients were selected and data were extracted (e.g. year 2022).

 -Table 1: please indicate the units of serum creatinine values

-Table 1: please indicate if any patient in a critical situation, in an intensive care unit, as this could have an association with the findings.

-Table 1: please, indicate in the first row that vancomycin levels correspond to STOOL samples

-Table 1 and 2: please, also provide estimated glomerular filtration rate o serum creatinine clearance values if possible

-Table 1 and 2: please indicate that vancomycin dose corresponds to the mg/kg PER ADMINISTRATION, and also provide the mean and SD or median and range of the total daily dose received in mg, since the dosage interval may be highly variable among the participants

-Lines 88-89 read as follows: “Two days of vancomycin were administered prior to stool collection with a vancomycin serum level of 17.4 mcg/mL on day 3”. However, this concentration value does not figure in table 2 for patient 2 (according to table 2 this value corresponds to patient 3). The concentration of 17.4 mcg/ml corresponded to day 3, although concentration data shown in table 2 corresponded to the period within 48 hours of stool collection. Please, indicate the same value both in the text and in table 2 or clarify the inconsistency.

-Line 93: I understand that patient 2 received IV vancomycin for at least 3 days, and line 93 reads that “antibiotics were administered for post-operation infection prophylaxis”. However, antibiotics in surgical prophylaxis are generally administered for 24 hours in total (1-2 doses), as recommended in clinical guidelines. Please confirm that the indication for IV vancomycin use in this patient was surgical prophylaxis.

-Line 98-99 read “The patient received 4 days of vancomycin prior to stool collection with serum levels of 6 mcg/mL and 17.1 mcg/mL on day 1 and day 4, respectively”. However, none of these values is indicated in table 2 for patient 3. According to table 2, the concentration of 6 mcg/mL corresponds to patient 2. Please, clarify this inconsistency.

-As indicated in table 2, two of the three patients were positive C. difficile and I suppose that they would be receiving some oral treatment (such as metronidazole or fidaxomicin) for treating it. I would suggest adding mention to the specific treatment that each of the patient was receiving to treat C. difficile.

-Line 116 reads the following: “Overall levels of vancomycin ranged from <2–94.8 mcg/mL”. Please specific that these values correspond to FECAL vancomycin levels.

-Lines 119-121 read the following “The overall rate of detectable fecal vancomycin in our study was comparable to previous reports: 9.1% (3/33) versus 19.2% (5/26) (21)”. Please, add more details of the characteristics of the study population referred to (study with reference number 21 in order to assess whether the population of this study is similar to that evaluated by the authors of the present manuscript authors (¿also had risk factors?) and to be able to properly interpret the information.

-Materials and methods: Please add mention to the statistics (mean, SD, median, range, etc.) and to the normality test and other statistical test used in the manuscript (if corresponds).

-Line 160 reads the following: “We report that >9% of patients with antibiotic-associated diarrhea have detectable…”. I suggest indicating a “9%” instead of more than a 9%. I also suggest adding in this sentence the mean and SD or the median and range of the vancomycin fecal levels detected.

Author Response

Reviewer #2

-Line 24 reads “Three of 33 (9%) samples had a detectable vancomycin concentration (range 1.2-13.2 mcg/mL)”. Please, indicate that these values correspond to FECAL samples and indicate the mean/median time to which these levels correspond with respect to the beginning of IV vancomycin treatment.

Updated Line 24 to read “Three of 33 (9%) stool samples had a detectable….” To clarify these are stool concentrations. The range is provided as only one stool sample per patient was taken at time of stool collection for C. difficile infection testing rather than serial samples.

 -Line 50: please indicate to which number of participants the samples correspond. I understand that there were 33 different patients, but this should be made explicit in the text.

To clarify, there was only one stool sample taken from each patient. This stool sample was taken as routine clinical care for C. difficile infection testing. No serial samples were taken after this initial sample. Our biorepository was screen for stool samples/patients that met the inclusion criteria pre-defined and those were the ones tested via HPLC.

-Lines 134-135: please add the information related to the setting from which patients were selected (medical, surgical or both), the age range of the participants (>18 years), and the study period in which patients were selected and data were extracted (e.g. year 2022).

Materials and Methods section have been updated with age range and study period from which the samples were collected. (line 135-137)

 -Table 1: please indicate the units of serum creatinine values

Included – thank you for the edit!

-Table 1: please indicate if any patient in a critical situation, in an intensive care unit, as this could have an association with the findings.

Thank you for the suggestion, Table 1 has been updated with ICU status at time of stool collection.

-Table 1: please, indicate in the first row that vancomycin levels correspond to STOOL samples

If referring to the heading of the table, the demographics are stratified by “Vancomycin detected in stool” vs “No vancomycin detected in stool” in its current state. The last line of the table “Vancomycin serum trough level” does indeed refer to the serum concentration as we wanted to see if there was correlation to excessive serum concentrations and stool concentrations. Will defer to the editor if this needs to be further clarified.

-Table 1 and 2: please, also provide estimated glomerular filtration rate o serum creatinine clearance values if possible

The authorship team had previously decided to only include serum creatinine and renal replacement status to exhibit the renal function status to the reader. This was decided due to CrCl not accurately representing patients receiving renal replacement and with the smaller numbers in this study, skewed the representation.

-Table 1 and 2: please indicate that vancomycin dose corresponds to the mg/kg PER ADMINISTRATION, and also provide the mean and SD or median and range of the total daily dose received in mg, since the dosage interval may be highly variable among the participants

Thank you for your comment on clarifying this was per administration; table has been updated to mg/kg/dose. In regards to the total daily dose, the authorship team decided to forgo including this information as several patients were on renal replacement specifically hemodialysis thus was dosed purely based on levels which would skew the mean/media for the cohort. We agree that the dosage interval can vary greatly, thus we included the mean vancomycin serum trough levels to attempt to let the reader access appropriateness of IV vancomycin regimen. Will defer to the editor if this needs further information or clarification.

-Lines 88-89 read as follows: “Two days of vancomycin were administered prior to stool collection with a vancomycin serum level of 17.4 mcg/mL on day 3”. However, this concentration value does not figure in table 2 for patient 2 (according to table 2 this value corresponds to patient 3). The concentration of 17.4 mcg/ml corresponded to day 3, although concentration data shown in table 2 corresponded to the period within 48 hours of stool collection. Please, indicate the same value both in the text and in table 2 or clarify the inconsistency.

Thank you for the edit. It was a transcription error in Table 2 and the text is correct. The authorship team appreciated your thoroughness.

-Line 93: I understand that patient 2 received IV vancomycin for at least 3 days, and line 93 reads that “antibiotics were administered for post-operation infection prophylaxis”. However, antibiotics in surgical prophylaxis are generally administered for 24 hours in total (1-2 doses), as recommended in clinical guidelines. Please confirm that the indication for IV vancomycin use in this patient was surgical prophylaxis.

Thank you for the comment. The authorship team agrees that the duration was longer than recommended, however with the retrospective nature of the study no stewardship was feasible. The authorship team confirms that was the clinical teams indication.

-Line 98-99 read “The patient received 4 days of vancomycin prior to stool collection with serum levels of 6 mcg/mL and 17.1 mcg/mL on day 1 and day 4, respectively”. However, none of these values is indicated in table 2 for patient 3. According to table 2, the concentration of 6 mcg/mL corresponds to patient 2. Please, clarify this inconsistency.

Has been updated with previous comment. Thank you for the catch.

-As indicated in table 2, two of the three patients were positive C. difficile and I suppose that they would be receiving some oral treatment (such as metronidazole or fidaxomicin) for treating it. I would suggest adding mention to the specific treatment that each of the patient was receiving to treat C. difficile.

As this was all prior to C. difficile treatment (the sample tested was the stool sample sent for CDI confirmation), the authorship team does not see the utility in including the CDI treatment as this was after stool collection. The authorship team confirms that all CDI treatment (PO vancomycin, metronidazole, and fidaxomicin) were not started prior to stool collection.

-Line 116 reads the following: “Overall levels of vancomycin ranged from <2–94.8 mcg/mL”. Please specific that these values correspond to FECAL vancomycin levels.

Updated to include “stool levels” language.

-Lines 119-121 read the following “The overall rate of detectable fecal vancomycin in our study was comparable to previous reports: 9.1% (3/33) versus 19.2% (5/26) (21)”. Please, add more details of the characteristics of the study population referred to (study with reference number 21 in order to assess whether the population of this study is similar to that evaluated by the authors of the present manuscript authors (¿also had risk factors?) and to be able to properly interpret the information.

Discussion expanded upon and updated.

-Materials and methods: Please add mention to the statistics (mean, SD, median, range, etc.) and to the normality test and other statistical test used in the manuscript (if corresponds).

Materials and methods updated to include section on statistics completed.

-Line 160 reads the following: “We report that >9% of patients with antibiotic-associated diarrhea have detectable…”. I suggest indicating a “9%” instead of more than a 9%. I also suggest adding in this sentence the mean and SD or the median and range of the vancomycin fecal levels detected.

Section updated to be 9% rather than >9%.

Reviewer 3 Report

Dear Authors

Your brief report manuscript entitled "Detectable Vancomycin Stool Concentrations in Hospitalized Patients with Diarrhea Given Intravenous Vancomycin" has significantly contribution to clinical data presentation in hospital-based studies.  Work is interesting and has potential for the scientific community. 

This brief study is based on patients admitted to the hospital, although only three patients have such Vancomycin detected out of thirty-three (33).

 I think it is a small finding but if possible research will have to be done on a large scale or take more sample size.

 As it is a Drug (Pharmacy) related topic and more in the clinical field, therefore, I think it should be considered in some more specific areas like clinical finding journals or parts.

Authors or research has already taken samples in between intervals (not possible to take every hour sample). If possible take more samples (>200).

Yes I think it is a brief note and the author tried to find out only three patients were reported.

 It should be added more references but this small finding is enough but the possibility remains there to increase references.
There are no figures and only two tables that describe the details information gathered by authors and analysis.

Author Response

Reviewer #3
Your brief report manuscript entitled "Detectable Vancomycin Stool Concentrations in Hospitalized Patients with Diarrhea Given Intravenous Vancomycin" has significantly contribution to clinical data presentation in hospital-based studies.  Work is interesting and has potential for the scientific community. 
This brief study is based on patients admitted to the hospital, although only three patients have such Vancomycin detected out of thirty-three (33).

 I think it is a small finding but if possible research will have to be done on a large scale or take more sample size.
 As it is a Drug (Pharmacy) related topic and more in the clinical field, therefore, I think it should be considered in some more specific areas like clinical finding journals or parts.
Authors or research has already taken samples in between intervals (not possible to take every hour sample). If possible take more samples (>200).
Yes I think it is a brief note and the author tried to find out only three patients were reported.
It should be added more references but this small finding is enough but the possibility remains there to increase references.
There are no figures and only two tables that describe the details information gathered by authors and analysis.

Reply: Thank you for the comments and thoughts. The authorship team agrees this is an exploratory, proof of concept study to add to the scientific community. Larger prospective studies would be ideal to complete taking serial levels. Future work to see the implication of IV vancomycin penetrating the GI such as antimicrobial resistance generation or dysbiosis of the microbiome would be of great interest.